# Role of Denosumab in Patients with Intermediate Spinal Instability Neoplastic Score (SINS)

**DOI:** 10.3390/cancers17091539

**Published:** 2025-05-01

**Authors:** JunYeop Lee, Bong-Soon Chang, Hyoungmin Kim, Sung Taeck Kim, Seonpyo Jang, Sam Yeol Chang

**Affiliations:** Department of Orthopedic Surgery, Seoul National University Hospital, 101 Daehangro, Jongrogu, Seoul 03080, Republic of Korea; 7a917@snuh.org (J.L.); bschang@snu.ac.kr (B.-S.C.); hmkhm@snu.ac.kr (H.K.); 7a913@snuh.org (S.T.K.); 7a912@snuh.org (S.J.)

**Keywords:** spinal metastasis, spinal instability neoplastic score, Hounsfield unit, denosumab

## Abstract

Due to the increasing incidence of cancers, the evaluation and management of vertebral metastases are crucial. However, impending instability in spinal metastases is challenging due to a lack of consensus. Though denosumab has emerged as a treatment option for patients with bone metastases of solid tumors, there is limited evidence on the clinical effect of denosumab in impending instability. In this retrospective study, we investigated the role of denosumab in spinal metastases. Denosumab reduced the need for surgery in patients with intermediate spinal instability neoplastic score (7–12 points) compared to the propensity-matched control. In addition, denosumab improved mechanical pain and induced sclerotic changes in osteolytic lesions in these patients.

## 1. Introduction

Cancer rates are increasing, and the World Health Organization (WHO) has estimated that by 2040, there will be 29.4 million cancer cases globally [1]. Furthermore, the quality of life and life expectancy of cancer patients are improving, which implies an increasing population living with solid tumors. According to a study by Ruben et al. in 2022, the incidence of spinal metastases in solid tumors is as high as 15.67%, and this rate increases to 30% in postmortem autopsy studies [2]. Among patients with spinal metastases, approximately 9.5% present with metastatic epidural spinal cord compression, and 12.6% present with pathological fractures in the vertebral body. Therefore, the evaluation and management of vertebral metastases are as crucial as the treatment of the primary cancer itself [3,4].

Spinal instability due to metastatic lesion can cause axial pain and exacerbate spinal cord compression. Unlike instability caused by high-energy trauma, spinal instability due to cancer exhibits different characteristics, such as variations in the patterns affecting bones and ligaments, healing potential, and neurologic symptoms. These differences highlight the need for distinct evaluation tools for spinal instability in cancer patients. In 2010, Fisher et al. introduced the Spine Instability Neoplastic Score (SINS) as an assessment tool for evaluating instability in spinal metastases [5]. The SINS consists of six components: location, pain, bone lesion, radiographic spinal alignment, vertebral body collapse, and posterior ligamentous complex (PLC) involvement. Each component is scored individually, and the total score ranges from 0 to 18. Scores of 0 to 6 denote stability and that nonoperative treatment recommended, whereas scores of 13 to 18 denote instability and that surgical stabilization is required. However, treatment guidelines for patients with intermediate SINS (7–12) remain unclear.

Previous studies have sought to examine the fate of impending instability and establish guidelines for treating patients with intermediate SINS [6,7,8,9]. In these patients, radiotherapy provided temporary improvement in pain and function, and 10–30% of cases required surgical interventions during follow-up. Therefore, efforts to reduce the rate of conversion to surgery in patients with intermediate SINS lesions are crucial for improving their quality of life.

Meanwhile, denosumab has emerged as an essential treatment option for patients with bone metastases of solid tumor [10]. Denosumab is a fully human monoclonal antibody that inactivates Receptor Activator of Nuclear Factor Kappa-B ligand (RANKL). By binding to RANKL, in a manner similar to native osteoprotegerin, it disrupts the bone destruction cycle associated with metastasized tumor cells. Since 2010, it has also been included in guidelines for preventing skeletal-related events (SREs) in patients with skeletal metastases [11]. However, there is limited evidence on the clinical effect of denosumab in patients with impending instability due to spinal metastasis. Therefore, this study was conducted under the hypothesis that denosumab can improve spinal stability and reduce the rate of conversion to surgical treatment in patients with impending instability (intermediate SINS).

## 2. Materials and Methods

### 2.1. Study Design and Participants

This is a retrospective study that reviewed a consecutive series of patients with spinal metastases and an intermediate SINS score (7–12 points), treated between January 2017 and December 2023 in a single cancer hospital. Only patients who were initially treated non-operatively for the intermediate SINS lesion and followed up for at least 6 months were included in the study. Exclusion criteria were as follows: (1) patients who received denosumab after spine surgery, (2) those with a history of prior spinal metastases surgery, and (3) those who underwent surgical treatment at the time of diagnosis of spinal metastasis. The current study was approved by the Institutional Review Board (approval number: H-2407-015-1549), and the requirement for informed consent was waived due to its retrospective nature.

### 2.2. Data Collection and Outcome Assessment

Demographic data including age, sex, and primary cancer type were collected from the electronic medical records. The patients were divided into two groups: the D-group, consisting of those treated with denosumab (120 mg every 4 weeks, subcutaneously) and the N-group, consisting of those who did not receive denosumab treatment. Denosumab has been reported to be effective in spinal metastases of breast cancer; therefore, in this study, it was primarily administered to patients with breast cancer. In patients with other primary tumors, denosumab was administered on a random basis.

The primary outcome of this study was the rate of conversion to surgery ratio during the follow-up in both groups. The causes for conversion to surgery, such as neurological deficit and pain aggravation, were also assessed. The secondary outcome was the change in the total score and the scores in each category of SINS from the diagnosis of metastatic lesion to the final follow-up. The third outcome was the change in the Hounsfield unit (HU) from the diagnosis of metastatic lesion to the final follow-up. The mean HU values before (preHU) and after (postHU) treatment were measured on axial image of computed tomography scans of affected vertebral body.

### 2.3. Propensity Matching and Statistical Analysis

Propensity score analysis was estimated in this study for the reduction of bias. Propensity score matching (PSM) was performed using the MatchIt package in R version 4.4.1 (Foundation for Statistical Computing, Vienna, Austria). Covariates potentially influencing the conversion to surgery—age, sex, and primary cancer—were selected for matching to address bias. Primary cancer type was converted into a variable using the revised Tokuhashi scoring system [12]. We also examined whether radiation therapy was conducted in the propensity score-matched groups and assessed the presence of epidural disease and paraspinal metastases using magnetic resonance imaging (MRI). We applied the nearest neighbor propensity score matching to construct comparable cohorts, thereby minimizing confounding effects from baseline characteristics. Although the overall sample size could be reduced, we performed 1:1 propensity score matching to minimize potential bias and achieve optimal covariate balance between groups. Adequate balance between the groups was assessed using standardized mean differences (SMDs) < 0.1 and a caliper width of 0.025 for the covariates.

For group comparisons, a chi-square test was used to compare categorical variables, and Student’s *t*-test was applied for continuous variables. The Kaplan–Meier survival analysis was used to evaluate the rate of conversion to surgery in both groups. A *p*-value < 0.05 was considered statistically significant. All statistical analyses were conducted using R version 4.3.3 (The R Foundation for Statistical Computing, Vienna, Austria).

## 3. Results

### 3.1. Characteristics of Study Participants

The current study included 286 patients (male: 151, female: 135) with a mean age of 68.0 ± 12.6 years and a mean follow-up period of 37.1 ± 26.5 months (Table 1). Forty-one (14.3%) patients received denosumab and were designated as the D-group, whereas 245 (85.7%) did not receive denosumab for their intermediate SINS lesions (N-group). There was a significantly higher proportion of females in the D-group than the N-group (75.6% vs. 42.4%, *p* = 0.001), primarily due to a higher rate of breast cancer patients in the D-group (43.9%). The most common primary cancer was lung cancer (n = 85), followed by breast cancer (n = 58) and hepatocellular carcinoma (n = 39). The average duration of denosumab treatment was 9.8 ± 10.7 months, and there were no cases of serious adverse events of denosumab, such as osteonecrosis of jaw and atypical femoral fractures, in this cohort.

After PSM and 1:1 nearest-neighbor matching with a caliper of 0.025, the mean propensity scores between the denosumab and control groups were nearly identical (0.2223 vs. 0.2219), indicating excellent covariate balance. In the PSM cohort, the D-group and N-group had 36 patients each, consisting of 10 male and 26 female patients. The mean age was 65.3 ± 14.0 and 67.0 ± 13.2 years, respectively (*p* = 0.588). According to the modified Tokuhashi classification, the primary cancer scores in the D-group were as follows: twenty-four patients scored five points, one patient scored three points, three patients scored two points, two patients scored one point, and six patients scored zero points. In the N-group, the corresponding distribution was twenty-three, four, two, two, and five patients, respectively (*p* = 0.715). In the N-group, five patients received bisphosphonate. Baseline vertebral compression fractures (VCFs) were seen in 22 patients in the D-group and 21 patients in the N-group. The baseline characteristics of the SINS component variables were compared between the denosumab group (D-group) and the non-denosumab group (N-group). The distributions of scores for location, pain, bone lesion, radiographic alignment, body collapse, and PLC involvement were generally similar between the two groups. For location, pain, and bone lesion, the score patterns were comparable, with no statistically significant differences (*p* = 0.173, *p* = 0.066, and *p* = 0.334, respectively). Radiographic alignment also showed a similar distribution, with most patients in both groups classified as score 0 (*p* = 0.643). Although a slight difference was observed in the body collapse component (*p* = 0.046), with a higher proportion of patients without collapse in the D-group, the overall distribution remained broadly consistent between groups. PLC involvement showed no significant difference either (*p* = 0.161), supporting that the baseline SINS component profiles were well balanced between the groups (Table 2). In the D-group, 50% of patients (18/36) received radiation therapy: fifteen patients with palliative radiation therapy and three with stereotactic radiosurgery (SRS). A total of 61.1% of patients (22/36) in the N-group received radiation therapy: nineteen patients with palliative radiation therapy and three with SRS. This difference was not statistically significant (*p* = 0.477). Epidural disease was noted in six cases in the D-group and eleven in the N-group (*p* = 0.267). Paraspinal metastases were equally observed in six cases in both groups. During the treatment, no severe adverse effects were observed in either the D-group or the N-group. Hypocalcemia occurred in 25.0% of patients (9/36) in the D-group and 36.1% (13/36) in the N-group, with no statistically significant difference between the groups (*p* = 0.443). This did not show a substantial difference from the reported incidence rate of denosumab-induced hypocalcemia in patients with solid tumors [13].

### 3.2. Comparison of Conversion-to-Surgery Rate

The rate of conversion to surgery was higher in the N-group (18.8%, 46/245) than the D-group (9.8%, 4/41), although the difference did not reach statistical significance in crosstab analysis (*p* = 0.159). There were four patients in the D-group who underwent surgical treatment for intermediate SINS lesion despite denosumab treatment: two patients due to pain aggravation and two due to neurological deficit. In the N-group, 46 patients underwent surgical treatment: 24 patients due to pain aggravation and 22 due to neurologic deficit. After PSM, the rate of conversion to surgery was twice as high in the N-group as in the D-group (8.3% [3/36] vs. 16.6% [6/36]). Although the odds of undergoing surgery were higher in Group A (OR = 2.20, 95% CI: 0.51–9.50), the difference was not statistically significant (*p* = 0.285). In the survival analysis using Kaplan–Meier curves, the D-group showed a significantly lower probability of conversion to surgery than the N-group before and after PSM (*p* = 0.015 and *p* = 0.023, respectively) (Figure 1).

### 3.3. Changes in SINS and HU

The changes in SINS for the two groups are described in Figure 2 and Table 3. The total SINS showed significant improvement in the D-group before and after PSM (*p* = 0.001), from 9.1 at baseline to 7.6 at the final follow-up before PSM, and from 9.1 to 7.5 after PSM. Among the categorical scores, pain (from 2.3 to 1.0, *p* = 0.001) and bone lesion (from 1.6 to 1.3, *p* = 0.048) scores significantly improved after denosumab treatment. As for the N-group, the total SINS decreased from 9.2 to 8.9 and from 8.8 to 8.1 before and after PSM, but the difference was not statistically significant (*p* = 0.669 and *p* = 0.058, respectively). The alignment and body collapse scores worsened, significantly increasing from 0.2 to 0.4 and from 1.3 to 1.6, respectively. Following propensity score matching, the D-group demonstrated an increase in mean HU from 202.6 to 378.1, whereas the N-group exhibited a slight decrease from 243.8 to 236.5 during treatment. This difference was statistically significant.

## 4. Discussion

As cancer incidence rises and patients with bone metastases live longer, metastatic disease commonly involves the spine [14]. Because instability due to spinal metastases can cause pain, deformity, and neurologic deficit, surgical stabilization is often considered in patients with impending instability. However, since surgical treatment for metastatic lesions carries surgical morbidity [15], and systemic cancer treatment may have to be ceased during surgery and recovery, it is critical to minimize unnecessary operations [16]. Denosumab can be a useful adjuvant treatment option to reduce the rate of conversion to surgery in patients with impending instability, which was initially treated nonoperatively.

In this study, we investigated the effect of denosumab on the conversion rate to surgery in patients with impending instability by comparing patients who received and did not receive denosumab after diagnosis of spinal metastasis with intermediate SINS. As for the total cohort, 9.8% (4/41) of patients in the D-group and 18.8% (46/245) of patients in the N-group underwent surgery for metastatic spine lesions during the follow-up period. Approximately twice as many patients in the N-group underwent surgical stabilization as in the D-group.

Because of the discrepancies in the proportions of primary cancers between the two groups (e.g., a significantly higher proportion of patients with breast cancer in the D-group), we performed PSM on age, sex, and primary cancer. After PSM, the difference in the surgical rate between the two groups persisted, with the rates of conversion to surgery being 8.3% (3/36) for the D-group and 16.6% (6/36) for the N-group. Additionally, survival analysis using Kaplan-Meier curves demonstrated that denosumab had a protective effect in reducing the need for surgery (Figure 1).

The changes in SINS during the follow-up period may explain the mechanism of reduction in the rate of surgical treatment in the D-group. Notably, the bone lesion score in the D-group significantly decreased from 1.6 to 1.3 (*p* = 0.048, SMD = 0.440), suggesting a transformation from an osteolytic lesion to a mixed osteoblastic and osteolytic lesion (Figure 3). Meanwhile, the alignment score in the N-group increased from 0.2 to 0.4 (*p* = 0.033, SMD = −0.0246), and the body collapse score also significantly increased from 1.3 to 1.6 (*p* = 0.004, SMD = −0.330). Denosumab reduces the need for surgical stabilization in patients with impending instability by improving osteolytic lesions and preventing vertebral collapse. Multiple previous studies have reported similar results in which denosumab reduced the incidence of SREs and delayed the onset of the first SRE event in patients with bone metastases [17,18,19].

Besides the anti-osteoclastic effect, an additional clinical effect of denosumab for patients with spinal metastasis is controlling pain arising from the metastatic lesions. In our study, the pain score showed significant reductions in both groups after various systemic and local treatments for metastatic lesion, such as chemotherapy and radiotherapy. Although both groups showed statistically significant improvements in the pain score, the D-group showed a more pronounced decrease (Table 3). Previous studies in the literature have shown the clinical effectiveness of denosumab for controlling pain in patients with skeletal metastasis [20,21,22]. Our analysis also shows similar results, indicating the effect of denosumab in patients experiencing pain from bone metastases.

The Hounsfield unit (HU) is a measurement used to evaluate the bone mineral density based on computed tomography (CT) and has been recommended for the quantitative evaluation of treatment responses in metastatic bone disease [23]. The change in HU in affected areas after treatment represents cellular changes and is a promising marker of treatment response. A previous study has reported HU measurements not only serve as an approach for evaluating treatment response, but also predict overall survival in cancer metastases [24]. In our study, the D-group exhibited a more favorable change in HU values than the N-group. We also tried to determine quantitative HU values statistically associated with clinical outcomes, such as the conversion to surgery ratio or changes in SINS. However, due to the small sample size, the cut-off value of delta HU could not be demonstrated. As the number of patients with spinal metastases treated with denosumab increases, further studies will be warranted.

Although radiotherapy is the primary treatment option for symptomatic spinal metastasis, vertebral compression fracture (VCF) following radiotherapy is a significant concern in patients with spinal metastasis. VCF following radiotherapy can lead to the need for surgical stabilization, especially in patients with osteolytic lesions and impending instability [25]. For such patients with a high risk of having VCF, prophylactic surgical stabilization is often considered before undergoing radiotherapy, especially when stereotactic radiosurgery (SRS) is planned. As an alternative to prophylactic surgical stabilization, denosumab can be considered to minimize the occurrence of VCF after radiotherapy. Future studies should examine the preventive effect of denosumab on VCF following radiotherapy.

This study has several limitations. First, as a single-center study, the current cohort has a relatively small sample size and heterogeneous primary cancers. Therefore, we performed propensity score analysis to account for factors that might affect treatment outcomes. Although age, sex, and primary cancer were included as covariates, the limited number of patients, especially in the D-group, restricted the inclusion of additional variables in the analysis. Second, because denosumab is a relatively recent advancement, the D-group primarily consists of later-phase patients with a shorter follow-up period. Third, due to the retrospective nature, the protocol of denosumab treatment was not standardized and strictly controlled. Well-designed prospective studies are required to further clarify the clinical and radiological effect of denosumab on patients with impending instability due to spinal metastasis. However, this is the first study to investigate the clinical impact of denosumab in patients with intermediate SINS.

## 5. Conclusions

In this study, denosumab lowered the rate of conversion to surgery in patients with impending instability (intermediate SINS) due to spinal metastasis. Patients who received denosumab treatment showed significant improvements in the total, pain, and bone lesion scores of SINS and the Hounsfield unit. Denosumab is a viable treatment option for patients with impending instability due to spinal metastasis.

## Figures and Tables

**Figure 1 cancers-17-01539-f001:**
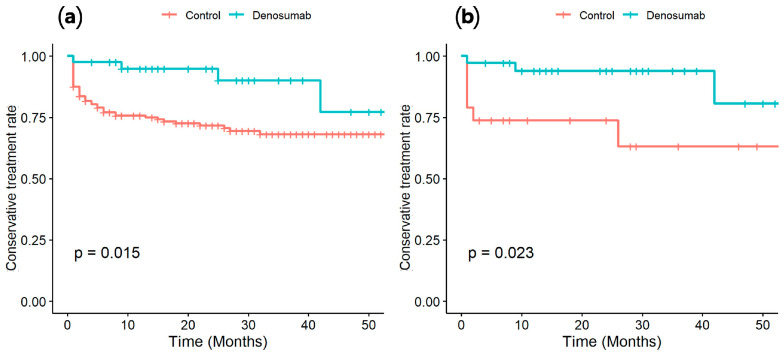
Results of Kaplan–Meier survival analysis of conversion to surgery before propensity score matching (**a**) and after (**b**).

**Figure 2 cancers-17-01539-f002:**
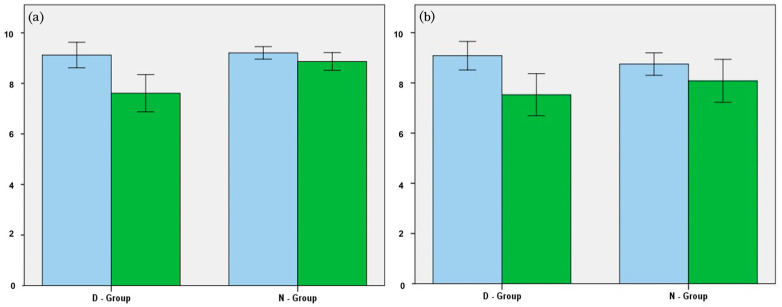
Change in spinal instability neoplastic score (SINS) before (**a**) and after (**b**) propensity score matching. The blue bars indicate SINS at the diagnosis of metastatic lesion, and the green bars indicate SINS at the final follow-up.

**Figure 3 cancers-17-01539-f003:**
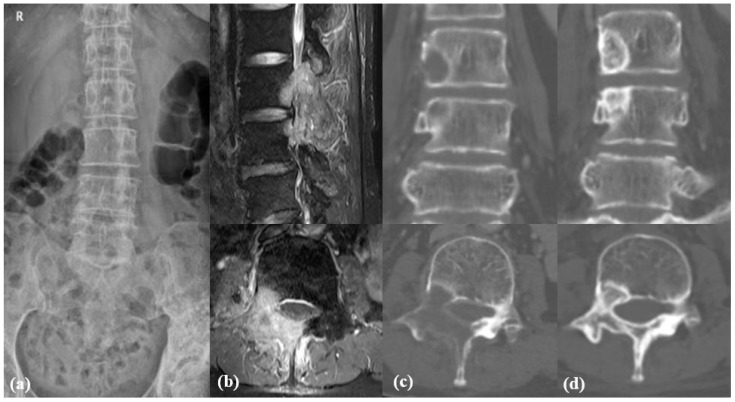
A 60-year-old female patient with breast cancer revealed osteolytic metastases at L3 and L4 in Radiographs (**a**), magnetic resonance imaging (**b**) and computed tomography (**c**). After 12 doses of denosumab, osteolytic bone lesions were converted to mixed lesions (**d**).

**Table 1 cancers-17-01539-t001:** Baseline characteristics of study participants.

Characteristics	Total (n = 286)	D-Group (n = 41)	N-Group (n = 245)	*p* Value
Age (means)	68.0 ± 12.6	65.2 ± 14.1	68.5 ± 12.3	0.128
Male–female	151:135	10:31	141:104	0.001
Follow-up (month)	37.1 ± 26.5	24.6 ± 15.1	39.2 ± 27.4	0.001
Primary cancers (n, %)				
Lung	85 (29.7%)	6 (14.6%)	79 (32.2%)	0.022
Breast	58 (20.3%)	18 (43.9%)	40 (16.3%)	0.001
Liver	39 (13.6%)	1 (2.4%)	38 (15.5%)	0.024
Kidney	24 (8.4%)	2 (4.9%)	22 (9.0%)	0.380
Prostate	24 (8.4%)	3 (7.3%)	21 (8.6%)	0.788
Sarcoma	23 (8.0%)	2 (4.9%)	21 (8.6%)	0.421
Thyroid	20 (7.0%)	7 (17.1%)	13 (5.3%)	0.006
Other	13 (4.6%)	2 (4.9%)	11 (4.5%)	0.912

**Table 2 cancers-17-01539-t002:** SINS (Spinal Instability Neoplastic Score) components score of PSM (propensity score matching) cohort.

	D-Group	N-Group	*p* Value
Location			0.173
1	6	13	
2	13	10	
3	17	13	
Pain			0.066
0	3	0	
1	7	3	
3	26	33	
Bone lesion			0.334
0	1	4	
1	13	10	
2	22	22	
Radiographic alignment			0.643
0	34	33	
2	2	3	
Body collapse			0.046
0	11	9	
1	4	11	
2	14	15	
3	7	1	
Posterolateral involvement			0.161
0	16	14	
1	7	14	
3	13	8	

Note: D-Group is denosumab-injected group, N-Group is non-injected group.

**Table 3 cancers-17-01539-t003:** Change in SINS (spinal instability neoplastic score) in the D-group and N-group.

	D-Group			N-Group		
	Initial	Final	*p*-Value	Initial	Final	*p*-Value
Location	2.3	2.3	1.000	2.1	2.1	1.000
Pain	2.3	1.0	0.000	2.7	2.0	0.000
Bone lesion	1.6	1.3	0.048	1.7	1.6	0.241
Alignment	0.1	0.2	0.649	0.2	0.4	0.033
Body collapse	1.4	1.5	0.768	1.3	1.6	0.004
Posterolateral involvement	1.3	1.3	1.000	1.1	1.1	0.669
Total SINS	9.1	7.6	0.001	9.2	8.9	0.669

## Data Availability

The data presented in this study are available on request from the corresponding author. The data are not publicly available due to privacy or ethical restrictions.

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
