# Peer review of "Role of Denosumab in Patients with Intermediate Spinal Instability Neoplastic Score (SINS)"

_cancers, 2025, doi:10.3390/cancers17091539_

Round 1
Reviewer 1 Report (Previous Reviewer 1)
Comments and Suggestions for Authors
The authors answered all questions of my review
Reviewer 2 Report (Previous Reviewer 2)
Comments and Suggestions for Authors
All comments have been addressed
This manuscript is a resubmission of an earlier submission. The following is a list of the peer review reports and author responses from that submission.
Round 1
Reviewer 1 Report
Comments and Suggestions for Authors
The authors present their matched-pair analysis on the role of Denosumab in patients with an intermediate Spinal Instability Neoplastic Score. Out of 286 patients, 41 received Denosumab while 245 did not; the reasons for the treatment decision remain unclear. The Denosumab group appears to have a better outcome. I have some questions that I would like to pose:
The rationale for why some patients received Denosumab while others did not is unclear. Even though this is a retrospective study, the authors should clarify the fundamental reasons for and against the therapy.
The labeling of Figure 1 needs to be revised. The Y-axis states "survival probability," but it actually represents the rate of conservatively treated patients without surgical intervention.
Did the authors consider quantifying the extent of osteolysis? For example, the number of vertebral bodies involved or which parts of the vertebral body were osteolytically altered? I believe it is likely that these factors influenced the treatment decision and are not represented in the currently used statistical model.
For Table 2, I suggest that alongside significance values, effect sizes (e.g., standardized mean differences) should also be included. However, these differences are sometimes marginal and clinically negligible.
Does the number of patients allow for a subgroup analysis to identify those who particularly benefited from Denosumab therapy?
It is quite surprising that no side effects were reported at all. This raises concerns about potential bias, suggesting that treatment side effects were not systematically recorded. This should at least be discussed and referenced against previous data on side effects.
What role did any (previous) radiation therapy play in this analysis?
The authors present their matched-pair analysis on the role of Denosumab in patients with an intermediate Spinal Instability Neoplastic Score. Out of 286 patients, 41 received Denosumab while 245 did not; the reasons for the treatment decision remain unclear. The Denosumab group appears to have a better outcome. I have some questions that I would like to pose:
The rationale for why some patients received Denosumab while others did not is unclear. Even though this is a retrospective study, the authors should clarify the fundamental reasons for and against the therapy.
> Author’s response: We appreciate the reviewer's comment. There was no clear indication for denosumab in our institution, and denosumab was prescribed individually based on the patient's clinical manifestations and radiological findings. However, there was a tendency for patients with breast and prostate cancer to receive denosumab treatment more than patients with other solid tumors because the national health insurance system in the author's region only covers denosumab for these patients. For this reason, we conducted propensity score matching (PSM) to minimize any confounding effect of the difference in primary cancer types between the two groups. We agree with the reviewer's comment that this is a limitation of a retrospective study.
The labeling of Figure 1 needs to be revised. The Y-axis states "survival probability," but it actually represents the rate of conservatively treated patients without surgical intervention.
> Author’s response: We appreciate your kind comment. In response to your comment, we have modified Figure 1 accordingly.
Did the authors consider quantifying the extent of osteolysis? For example, the number of vertebral bodies involved or which parts of the vertebral body were osteolytically altered? I believe it is likely that these factors influenced the treatment decision and are not represented in the currently used statistical model.
> Author’s response: Thank you for the reviewer’s comment. We did not consider the number of involved vertebral bodies and therefore did not include it in the manuscript. However, we performed an additional analysis on this aspect. In the D-group, spinal metastases involving more than two vertebral bodies were observed in 72.2% of patients, whereas in the N-group, such findings were noted in 41.7% of patients(p=0.017). In the SINS subcategory for location, the proportion of patients with scores of 3 or 2 was 82.9% in the D-group and 69.0% in the N-group, indicating that denosumab tended to be administered more frequently in junctional or mobile segments. However, this difference was not statistically significant(p=0.093). This may reflect our efforts to minimize differences in metastatic locations, which could influence the decision between surgical and nonsurgical treatment approaches.
For Table 2, I suggest that alongside significance values, effect sizes (e.g., standardized mean differences) should also be included. However, these differences are sometimes marginal and clinically negligible.
> Author’s response: We appreciate your kind comment. In response to your comment, we have modified Table 3 accordingly.
Does the number of patients allow for a subgroup analysis to identify those who particularly benefited from Denosumab therapy?
> Author’s response: We appreciate your kind comment. Subgroup analyses were performed within the propensity score–matched cohort. We compared patients who underwent conversion to surgery with those who did not. All patients in the conversion group were male; however, given the small number of cases (n = 3), it is difficult to draw statistically meaningful conclusions. Additional subgroup analyses based on age, primary cancer type, radiation therapy, and SINS were also conducted, but no statistically significant differences were observed. Further research with a larger dataset will be warranted.
It is quite surprising that no side effects were reported at all. This raises concerns about potential bias, suggesting that treatment side effects were not systematically recorded. This should at least be discussed and referenced against previous data on side effects.
> Author’s response: We think that it is a very crucial point. In our study, we primarily focused on assessing the presence of osteonecrosis of jaw or atypical femur fracture among the known side effects of denosumab. As suggested by the reviewer, we additionally investigated treatment-related side effects. Among the PSM cohort, hypocalcemia was observed in 9 patients in the D-group and 13 patients in the N-group but there were no severe hypocalcemia(<6mg/dl). The incidence of hypocalcemia was consistent with previously reported rates in the literature (Reference : Nakamura, K.; Kaya, M.; Yanagisawa, Y.; Yamamoto, K.; Takayashiki, N.; Ukita, H.; Nagura, M.; Sugiue, K.; Kitajima, M.; Hirano, K.; et al. Denosumab-induced hypocalcemia in patients with solid tumors and renal dysfunction: a multicenter, retrospective, observational study. BMC Cancer 2024, 24, 218, doi:10.1186/s12885-024-11942-2.). These findings have been added to the Results section of the manuscript(page 5).
What role did any (previous) radiation therapy play in this analysis?
> Author’s response: We appreciate your kind comment. In this study, we analyzed radiation therapy within the PSM cohort to account for potential confounding factors that may influence the progression of spinal metastases aside from denosumab. Radiation therapy was administered to 50.0% of patients in the D-group and 61.1% in the N-group, with no statistically significant difference between the two groups. These findings have been added to the Results section of the manuscript(page 5).
Reviewer 2 Report
Comments and Suggestions for Authors
Numerous very important baseline characteristics are missing.
?Bisphosphonate use
?Radiation use - palliative vs. SRS
?Epidural disease
?Paraspinal mass
?Individual SINS categories (including baseline VCF) in Table 1
PS matching on a very limited set of variables (age, sex, primary site) was unlikely to result in a fully balanced cohort
PS methods and results were poorly described. Refer to Yao et al JNCI 2017 for details.
The authors presented reasons for proceeding to surgery in the denosumab group but curiously did not report this for the non-denosumab group
The authors stated that radiotherapy was 'examined' in the methods but the results were curiously never presented or mentioned again
Numerous very important baseline characteristics are missing.
?Bisphosphonate use
> Author’s response: We appreciate your kind comment. Our data collection for this aspect was insufficient and subsequently collected additional data. Within the PSM cohort, a total of five patients received bisphosphonate treatment. This information has been added to the Results section (page 4) of the manuscript.
?Radiation use - palliative vs. SRS
> Author’s response: Thank you for your comment. Additional data collection was conducted, and 18 patients in the D-group received radiation therapy. Among them, 41.7% (15/36) received palliative therapy and 8.3% (3/36) underwent stereotactic radiosurgery (SRS). In the N-group, 52.8% (19/36) received palliative therapy and 8.3% (3/36) received SRS. Statistically significant differences weren’t observed between the two groups(p=0.477). This information has been added to the Results section (page 5) of the manuscript.
?Epidural disease
> Author’s response: Thank you for your comment. Additional investigation was conducted regarding the presence of epidural disease. In the D-group, 6 patients were identified as having epidural involvement, while 11 patients in the N-group showed similar findings. This difference was not statistically significant between the two groups. We added to the Results section (page 5) of the manuscript.
?Paraspinal mass
> Author’s response: Thank you for your comment. We also examined the presence of paraspinal invasion. 6 patients in the D-group and 6 patients in the N-group were identified as having paraspinal invasion. We added to the Results section (page 5) of the manuscript.
?Individual SINS categories (including baseline VCF) in Table 1
> Author’s response: Thank you for your comment. We also examined about reviewer’s comments. In the PSM cohort, we investigated the distribution of individual SINS categories and the baseline incidence of vertebral compression fractures (VCFs). As shown in the table 2, no statistically significant differences were observed between the two groups in any SINS subcategory, except for the vertebral body collapse score. The proportion of patients with baseline compression fractures was also comparable between the groups. We added to the Results section (page 4,5) of the manuscript.
PS matching on a very limited set of variables (age, sex, primary site) was unlikely to result in a fully balanced cohort
PS methods and results were poorly described. Refer to Yao et al JNCI 2017 for details.
> Author’s response: We appreciate your kind comment. Among the potential confounding variables that could have influenced the results of this study, age, sex, and primary cancer type were considered the most important. While additional covariates may exist, as mentioned in the limitations(page 8), the relatively small sample size limited our ability to include multiple variables in the model without compromising the statistical power of the analysis. We acknowledge that several important aspects of propensity score (PS) analysis had been previously overlooked in our initial submission. Thanks to the reviewer’s insightful comments, we were able to revise the manuscript into a more rigorous and higher-quality version.
Specifically, we have added a detailed description of the PS analysis methodology in the Materials and Methods section (Page 6), clearly outlining how the matching was performed. In the Results section (Page 4), we further evaluated the overlap of propensity score distributions between the matched groups to assess the adequacy of the matching and confirmed that the distribution of covariates was well balanced as intended.
Additionally, we included the calculation of the confidence interval for comparison the rate of conversion to surgery in two groups(Page 6).
The authors presented reasons for proceeding to surgery in the denosumab group but curiously did not report this for the non-denosumab group
> Author’s response: We think that it is a very crucial point. As the primary objective of this study was to evaluate whether denosumab could reduce the conversion rate to surgery in patients with spinal metastases, we did not initially investigate the reasons for proceeding to surgery in the non-denosumab group. However, in response to the reviewer’s comment, we conducted additional data collection. Among the 46 patients in the N-group who underwent surgery, 24 patients underwent surgery due to pain, and 22 patients due to neurologic deficits. We added to the Results section (page 5) of the manuscript.
The authors stated that radiotherapy was 'examined' in the methods but the results were curiously never presented or mentioned again
> Author’s response: Thank you for your comment. As we considered that radiation therapy could influence the decision-making process regarding conversion to surgery, we examined whether radiation therapy was administered during the follow-up period. As mentioned above, 18 out of 36 patients in the D-group (50.0%) and 22 out of 36 patients in the N-group (61.1%) received radiation therapy. This difference was not statistically significant.
On behalf of all authors, I would like to once again thank the reviewer for a comprehensive review and constructive comments, which were most valuable to us in preparing a more balanced and improved manuscript.